# Surgery Versus Chemoradiation Therapy for Oropharyngeal Squamous Cell Carcinoma: A Multidimensional Cross-Sectional Study

**DOI:** 10.3390/diseases13040106

**Published:** 2025-04-02

**Authors:** Giuseppe Riva, Dario Gamba, Simone Moglio, Giuseppe Carlo Iorio, Chiara Cavallin, Umberto Ricardi, Mario Airoldi, Andrea Canale, Andrea Albera, Giancarlo Pecorari

**Affiliations:** 1Division of Otorhinolaryngology, Department of Surgical Sciences, University of Turin, Via Genova 3, 10126 Turin, Italy; dario.gamba@unito.it (D.G.); simone.moglio@unito.it (S.M.); andrea.canale@unito.it (A.C.); aalbera@hotmail.com (A.A.); giancarlo.pecorari@unito.it (G.P.); 2Division of Radiation Oncology, Department of Oncology, University of Turin, Via Genova 3, 10126 Turin, Italy; giuseppecarlo.iorio@libero.it (G.C.I.); ccavallin@cittadellasalute.to.it (C.C.); umberto.ricardi@unito.it (U.R.); 3Division of Medical Oncology, Department of Oncology, University of Turin, Via Genova 3, 10126 Turin, Italy; mairoldi@cittadellasalute.to.it

**Keywords:** oropharyngeal cancer, oropharyngeal squamous cell carcinoma, chemoradiation, surgery, quality of life, sleep, psychological distress

## Abstract

Background/Objectives: The management of oropharyngeal squamous cell carcinoma (OPSCC) often involves multidisciplinary decision-making to optimize patient outcomes. Surgery and chemoradiation therapy (CRT) represent the two main treatment modalities. The aim of this cross-sectional study was to provide a comprehensive analysis of quality of life, speech, swallowing, sleep, psychological distress, and nutritional status in OPSCC patients treated with either surgery or CRT. Methods: Thirty subjects were divided into two groups based on treatment modality (>12-month follow-up): (A) surgery ± adjuvant treatment (15 patients); (B) exclusive CRT (15 patients). A multidimensional evaluation was performed by means of validated questionnaires. The following parameters were analyzed: quality of life, speech, swallowing, sleep quality, risk of sleep apnea, sleepiness, psychological distress, pain, and nutritional status. Results: No statistically significant difference was found between the two study groups for every parameter. The EORTC QLQ-C30 globally showed a good quality of life in both groups. Poor sleep quality was observed in 9 (60%) subjects in group A and in 6 (40%) patients in group B, respectively. Low, intermediate and high risk of malnutrition was observed in 73.3%, 20.0% and 6.7% of cases in group A, and in 93.3%, 6.7% and 0.0% in group B, respectively. Conclusions: Surgery and exclusive chemoradiotherapy appear to yield similar long-term outcomes across all evaluated dimensions, including quality of life, speech, swallowing, sleep, psychological distress, and nutritional status.

## 1. Introduction

Oropharyngeal squamous cell carcinoma (OPSCC) represents a growing public health concern, driven in part by the rising incidence of human papillomavirus (HPV)-associated cases [1]. The oropharynx is a critical region for vital functions such as swallowing, speech, and nutrition, as well as for overall quality of life (QoL). Treatment options for OPSCC typically aim to balance oncologic control with preservation of function. Among these, surgery combined with radiotherapy (SRT) and exclusive chemoradiotherapy (CRT) are two widely utilized approaches, each with distinct impacts on functional and QoL outcomes [2].

The management of OPSCC often involves multidisciplinary decision-making to optimize patient outcomes. Surgical approaches, particularly minimally invasive techniques like transoral robotic surgery (TORS), have gained traction due to their potential to reduce treatment-related morbidity [3]. Surgery is frequently followed by adjuvant radiotherapy (RT) or CRT in patients with high-risk pathological features, such as extracapsular spread or positive margins [4]. Conversely, CRT, which combines intensity-modulated radiotherapy (IMRT) with concurrent chemotherapy, is a non-surgical alternative that achieves comparable survival outcomes in many cases [5].

While these treatments are effective in controlling the disease, their impact on patient functionality and QoL often differs significantly. SRT may lead to higher rates of surgical complications but less radiation-induced toxicity, while CRT can preserve anatomy at the cost of increased long-term toxicities like xerostomia and dysphagia [6]. Understanding how these modalities affect speech, swallowing, nutrition, and overall QoL is essential for tailoring treatment plans to individual patient needs.

Quality of life is a multidimensional concept encompassing physical, emotional, and social well-being. In OPSCC, treatment often impacts QoL domains such as swallowing, speech, and social interaction. Surgical approaches, especially TORS, are associated with improved short- and long-term QoL outcomes compared to CRT in many cases. A systematic review found that patients undergoing surgery followed by radiotherapy experienced better QoL in domains related to saliva production and swallowing function than those treated with CRT alone [7].

CRT, on the other hand, is associated with significant acute and late toxicities that can severely affect QoL. Xerostomia, fibrosis, and swallowing difficulties are common, with many patients requiring prolonged gastrostomy tube dependence [8]. Despite these challenges, CRT remains a preferred option for advanced cases due to its ability to preserve anatomical structures and avoid surgical morbidity.

Speech and swallowing functions are particularly vulnerable to treatment-related damage in OPSCC. These functions are critical not only for nutrition but also for social interaction and overall patient dignity. Studies have shown that patients treated with CRT often experience better short-term speech outcomes compared to those undergoing surgery plus radiotherapy, due to preserved anatomical integrity [9]. However, CRT patients may face persistent challenges with swallowing due to radiation-induced fibrosis and mucositis.

Surgical interventions, particularly TORS, have been shown to improve long-term swallowing outcomes compared to CRT. A matched-pair analysis revealed that surgical patients had lower rates of gastrostomy tube dependence and improved swallowing function at two years post-treatment. However, the addition of adjuvant therapy in SRT can exacerbate swallowing difficulties, underscoring the importance of individualized treatment planning [10].

Nutritional status is another critical outcome for OPSCC patients, as treatment often results in significant weight loss, difficulty eating, and malnutrition. CRT patients frequently experience prolonged reliance on gastrostomy tubes, with up to 20% requiring them one year post-treatment [6]. Surgical patients may also face nutritional challenges, particularly if reconstructive procedures or adjuvant therapies are required, but they generally demonstrate quicker recovery in terms of oral intake and weight stabilization.

Despite advancements in treatment, direct comparisons between SRT and CRT in terms of QoL, speech, swallowing, and nutritional outcomes remain underexplored. Most studies are retrospective, with limited randomized data to guide clinical decision-making. For example, the PATHOS and ECOG-E3311 trials are currently investigating de-escalation strategies in SRT to improve functional outcomes, but their results are still pending [4].

This study aims to fill these gaps by providing a comprehensive analysis of QoL, speech, swallowing, sleep, psychological distress, and nutritional status in OPSCC patients treated with either surgery or CRT. By identifying the strengths and limitations of each approach, the findings will contribute to the development of more patient-centered treatment protocols.

## 2. Materials and Methods

### 2.1. Study Design

Thirty patients treated for OPSCC at our department were enrolled in this study. Exclusion criteria included age under 18, follow-up shorter than 12 months, presence of neurological and/or psychiatric conditions potentially affecting questionnaire compliance, and ongoing treatment for recurrence or secondary tumors. Only patients with over 12 months of follow-up were included to ensure side effects had stabilized. All procedures adhered to the ethical standards of the institutional research committee and the 1964 Helsinki Declaration with its amendments or equivalent ethical guidelines. Written informed consent was collected in every case. Approval was granted by the Institutional Review Board (A.O.U. Città della Salute e della Scienza di Torino-A.O. Ordine Mauriziano-A.S.L. Città di Torino).

Clinical data and treatment modalities were documented. Tumors were staged using the 8th edition of the AJCC TNM classification. Treatment followed both national and international protocols. Patients were categorized into two groups by treatment type: (A) surgery with or without adjuvant therapy (15 patients); (B) exclusive chemoradiotherapy (15 patients). TORS was used for tongue base lesions.

### 2.2. Procedures

Quality of life (EORTC QLQ-C30 and EORTC H&N35), sleep quality (Pittsburgh Sleep Quality Index—PSQI), sleep apnea risk (STOP-BANG), daytime sleepiness (Epworth Sleepiness Scale—ESS), psychological distress (Distress Thermometer—DT, Hospital Anxiety and Depression Scale—HADS), swallowing ability (MD Anderson Dysphagia Inventory—MDADI), and speech (Speech Handicap Index—SHI) were evaluated. Pain was assessed using a Visual Analog Scale (VAS, 0 = no pain; 10 = worst pain), and nutritional status via the Malnutrition Universal Screening Tool (MUST).

The EORTC-QLQ C30 includes 35 items assessing physical, role, emotional, social, and cognitive function and somatic symptoms, along with overall health. The specific module QLQ-H&N35 complements it with 35 items focused on symptoms specific to head and neck cancer. Higher symptom scores indicate more severe symptoms; higher global and functional scores indicate better quality of life [11,12].

The 19-item PSQI evaluated sleep over the previous month across seven domains: sleep quality, latency, duration, efficiency, disturbances, medication use, and daytime dysfunction. A total score above 5 denoted poor sleep quality [13,14].

Sleep apnea risk was measured with the 8-item STOP-BANG scale (yes/no) covering snoring, tiredness, observed apnea, hypertension, BMI > 35, age > 50, neck circumference > 40 cm, and male sex. Total score classified patients into low, moderate, or high risk [15].

ESS evaluated sleepiness with 8 scenarios rated 0–3 (0 = never; 3 = high), for a total of 0–24. Scores above 10 suggested excessive daytime sleepiness [16].

The DT was used to measure psychological distress on a scale from 0 (none) to 10 (extreme), referring to the previous week [17]. The HADS included 14 items (7 anxiety, 7 depression), each rated 0–3. Subscale scores ranged from 0 to 21 [18].

MDADI featured 20 items on a 5-point scale assessing swallowing difficulties. A total score was computed, where higher values reflected fewer problems [19]. SHI had 30 items on a 5-point scale assessing speech, with higher total scores denoting worse speech-related quality of life [20].

The MUST identified malnutrition risk based on BMI, weight loss, and impact of acute conditions [21].

### 2.3. Statistical Analyses

Statistical analysis was conducted using SPSS version 26.0. Since the Kolmogorov–Smirnov test indicated non-normal variable distribution, non-parametric methods were applied. A descriptive analysis was performed, with data presented as medians and interquartile ranges (IQR), or as percentages. Group comparisons used the Mann–Whitney U test for continuous variables and the Chi-square or Fisher’s exact test for categorical variables. A *p*-value below 0.05 was considered statistically significant.

## 3. Results

The median age of the entire sample was 62.5 years (IQR 11 years). The median Karnofsky Performance Status was 100% (as only three subjects presented a score of 80% and only eight presented a score of 90%; all the others reported a 100% score). The average mFI (modified Frailty Index, used to synthetize the comorbidities burden) was 0.063. The median follow-up was 25 months (IQR 36 months) for group A and 26 months (IQR 62 months) for group B (*p* = 0.838), respectively. No feeding tube or tracheotomy were present at evaluation. The clinical characteristics of the subjects included in both groups are reported in Table 1. Adjuvant RT was performed in 9 out of 15 patients in group A.

The median VAS value for pain was 0 (IQR 5) for group A and 0 (IQR 1) for group B (*p* = 0.389), while median BMI was 23.62 (IQR 6.20) for group A and 24.09 (IQR 6.69) for group B (*p* = 0.367), respectively. The EORTC QLQ-C30 globally showed a good quality of life in both groups (Table 2).

The sleep apnea risk evaluated by means of STOP-BANG was low in 10 (66.7%) subjects, intermediate in 3 (20.0%) patients, and high in 2 (13.3%) cases in group A, while it was low in 7 (46.7%) subjects, intermediate in 3 (20.0%) patients, and high in 5 (33.3%) cases in group B (*p* = 0.404). Table 3 reports the results about sleepiness (ESS) and sleep quality (PSQI). In particular, a poor sleep quality was observed in nine (60%) subjects in group A and in six (40%) patients in group B, respectively (*p* = 0.273).

The MUST identified a low, intermediate and high risk of malnutrition in 11 (73.3%), 3 (20.0%) and 1 (6.7%) patient in group A, and in 14 (93.3%), 1 (6.7%) and 0 (0.0%) patients in group B, respectively (*p* = 0.307). Table 4 highlights the results of swallowing (MDADI), speech (SHI), and psychological distress (DT and HADS).

The main results from this multidimensional evaluation of patients who underwent surgery or exclusive CRT for OPSCC are shown in Figure 1 and Figure 2.

## 4. Discussion

Oropharyngeal squamous cell carcinoma (OPSCC) is a heterogeneous disease with treatment strategies influenced significantly by tumor stage, location, and human papillomavirus (HPV) status. Advances in surgical techniques, radiotherapy, and systemic therapies have expanded the range of treatment options, necessitating tailored approaches based on disease stage to optimize outcomes and minimize treatment-related morbidity [2,22,23].

For early-stage OPSCC (stages I and II), the treatment aims to achieve local control while preserving function. Transoral robotic surgery (TORS), especially for tongue base cancer, is preferred for its low morbidity and comparable outcomes to radiotherapy, with fewer long-term toxicities [5]. Alternatively, intensity-modulated radiotherapy (IMRT) offers similar control but risks late complications like xerostomia and dysphagia [6].

In locally advanced OPSCC (stages III and IVa), treatment becomes more complex, often requiring multimodal approaches [24]. Surgery with adjuvant radiotherapy or CRT is a common strategy for resectable tumors, particularly when high-risk features such as extracapsular spread or positive margins are present. However, CRT as a standalone treatment is widely used for unresectable or borderline resectable cases and is supported by robust evidence demonstrating high survival rates, particularly in HPV-positive patients [25].

For advanced stages (IVb and IVc), systemic therapies play a dominant role. Chemotherapy combined with targeted therapies, such as cetuximab, alongside palliative radiotherapy, is frequently employed to manage symptoms and improve QoL. Immunotherapy, including checkpoint inhibitors, has shown promise in recurrent or metastatic settings, particularly for HPV-positive cases with a favorable tumor microenvironment [26].

Despite significant advancements, the choice of treatment modality for each stage is influenced by patient factors, tumor characteristics, and treatment toxicity profiles. Tailoring therapy based on tumor stage remains critical to balancing oncologic control and functional outcomes, emphasizing the need for continued research and innovation to optimize patient care.

Our study aimed to evaluate and compare the QoL, speech, swallowing, sleep, psychological distress, and nutritional status of OPSCC patients treated with surgery or exclusive CRT. Using multidimensional tools, the study provided comprehensive insights into the implications of each treatment modality.

While there were no significant differences found between the two study groups, we found some interesting trends that, although not statistically significant, are relevant and in line with clinical and literature findings.

Tumor sites were almost equally divided between the two groups with the exception of the soft palate tumors that were present only in the surgical group. In the latter group, most tumors were in early stages (53.3% stage I and 13.3% in stage II), while in the CRT group, the stages were more evenly separated with more patients affected by stage IV cancer (40%). The demographics and lifestyle factors of our patient cohort were in line with the epidemiology of this type of cancer and other similar studies, further affirming the validity of our sample as representative of the head and neck cancer population [27].

Our study showed that both groups reported generally good global health status as per the EORTC QLQ-C30, with no significant differences between surgery and CRT. These findings align with prior studies that have shown comparable QoL outcomes between surgical and non-surgical treatment modalities in OPSCC patients. For example, Roets et al. reported that CRT, while effective for oncologic control, was associated with significant acute and late toxicities, including xerostomia and dysphagia, which could negatively affect QoL [6]. Conversely, patients undergoing SRT, particularly with minimally invasive surgical techniques, often exhibited better saliva-related QoL. This was partially reflected in the current study, where issues like dry mouth were more common in CRT patients, although not statistically significant.

STOP-BANG and PSQI tools indicated similar sleep outcomes between the groups, further suggesting that neither modality had a dominant impact on sleep quality or disturbances. Worse quality of sleep, expressed by insomnia score at the EORTC questionnaire, was found in the surgical group compared to exclusive CRT, while less risk of obstructive sleep apnea syndrome (OSAS) was found in the same group, with a lower STOP-BANG score. Furthermore, there was no statistically significant difference between the two groups in the PSQI scores; this could be due to the fact the median scores were close to the cut-off (5 points). We can conclude that, at least in our little sample size, insomnia seemed to play a much bigger role than OSAS, and this was also present if we took into consideration the sleepiness (ESS), the PSQI-sleep disturbance and PSQI-daytime dysfunction items.

This contrasts with prior reports that linked CRT with higher risks of insomnia due to radiation-induced mucositis and chronic pain. However, our findings lacked statistical significance due to a low sample size between the two groups, so further investigation should be performed on this particular aspect.

Swallowing function is a critical aspect of post-treatment recovery, significantly influencing nutritional status and QoL. The study found no significant difference in MDADI scores between surgery and CRT groups, although CRT patients exhibited a trend toward better swallowing outcomes. This result aligns with findings from Gillespie et al., who suggested that CRT preserved anatomical structures and thus enhanced swallowing-related outcomes compared to surgical approaches [9]. However, the long-term impact of CRT-related fibrosis and tissue stiffness may negate these initial advantages. In a study by Ling et al., surgical patients showed improved swallowing function in the long term, with fewer requiring gastrostomy tube placement at two years post-treatment [10]. The mild swallowing impairments observed in the surgical group in the current study may reflect the burden of postoperative radiotherapy, which exacerbated swallowing difficulties due to mucosal damage and fibrosis.

Speech performance is another vital determinant of social integration and psychological well-being in OPSCC survivors. The Speech Handicap Index (SHI) showed no statistically significant differences between the groups, but CRT patients had lower median SHI scores, indicating potentially better speech outcomes. This trend aligns with findings from Yeh et al., who reported that CRT preserved speech function better than surgical approaches due to the avoidance of anatomical disruption [5]. However, surgical advancements, such as TORS, have reduced the impact of surgery on speech by minimizing damage to critical structures. Postoperative radiotherapy is also a key factor contributing to speech impairments in surgical patients. Lastly, this could also be due to the fact that the four patients with soft palate cancer were only in the surgical group. The surgery for this type of cancer has a high risk of rhinolalia which aggravates the speech perception and thus the overall QoL. Further studies are needed to clarify the interplay between surgery, adjuvant therapy, and speech outcomes.

Nutritional status, assessed using BMI and MUST, showed no significant differences between the groups in this study. However, CRT patients exhibited fewer intermediate and high-risk malnutrition cases. This result is surprising given the high prevalence of feeding tube dependence and mucositis-related dietary restrictions in CRT patients reported in prior studies [6,28]. However, this finding can be explained by the fact that our questionnaire is administered more than 12 months after the end of treatment where all the major complications, such as the one described above, have already been resolved. Surgical patients, despite undergoing potentially invasive procedures, generally recover oral intake faster, which may explain the comparable nutritional outcomes in the current cohort. Additionally, the use of minimally invasive techniques such as TORS may have mitigated the impact of surgery on nutrition. This finding is also present if we look at pain sores. VAS was reported to be 0 in both groups, signaling good recovery after treatment and an overall good outcome regarding pain.

Psychological outcomes, including anxiety and depression assessed through HADS, were similar between groups. This finding highlights the resilience of OPSCC patients irrespective of treatment modality. Emotional functioning scores from EORTC QLQ-C30 also reflected these trends. Social outcomes, including role functioning and social contact, were preserved in both groups. This is consistent with prior research indicating that, despite treatment-related challenges, OPSCC survivors often adapt well to their post-treatment lives [4].

This study highlights important strengths, making a meaningful contribution to understanding the comparative outcomes of surgery and exclusive CRT in OPSCC. The use of validated multidimensional tools ensured a comprehensive assessment of QoL, speech, swallowing, and nutritional outcomes. By directly comparing these two widely used treatment modalities, the study provides valuable insights into their relative benefits, focusing on clinically relevant, patient-centered outcomes. Additionally, the inclusion of a cohort representative of real-world OPSCC cases enhanced the generalizability of the findings.

This study has some limitations. The small sample size reduced statistical power and limited the ability to detect subtle differences between groups. RT may be a better option for patients who cannot undergo major surgery due to their overall condition and this may represent a bias for the study. Assessments conducted over 12 months post-treatment overlooked early complications, while the lack of instrumental evaluations for sleep disturbances, like OSAS, limited precision of sleep analysis. The absence of longitudinal data further constrains the ability to monitor functional recovery over time. Larger, prospective studies with stratified cohorts and pre- and post-treatment assessments are necessary to refine these findings and guide more tailored treatment strategies for OPSCC patients. Moreover, future studies should stratify for tumor stage.

## 5. Conclusions

Surgery combined with radiotherapy (SRT) and exclusive chemoradiotherapy (CRT) appear to yield similar long-term outcomes across all evaluated dimensions, including quality of life, speech, swallowing, sleep, psychological distress, and nutritional status. However, specific trends observed in this analysis warrant further investigation in future research. These trends may provide deeper insights into subtle differences between the two treatment modalities, especially in terms of functional recovery and patient-reported outcomes.

To build on these findings, future studies should include larger patient cohorts that allow for more robust stratification by tumor stage and anatomical site. Such stratification would help clarify the role of these variables in influencing treatment outcomes and guide more tailored therapeutic strategies. Additionally, future research should integrate instrumental evaluations for OSAS and employ more targeted questionnaires for insomnia. This approach could enhance the precision of assessing sleep-related outcomes, a critical component of QoL in OPSCC patients. Moreover, prospective studies comparing pre- and post-treatment outcomes are essential to better understand the trajectory of recovery and functional changes over time. This design would provide valuable longitudinal data on the direct impact of surgery and CRT on patient functionality and QoL, supporting more informed decision-making in clinical practice.

In conclusion, while surgery and CRT are both effective in managing OPSCC, further exploration through well-designed studies is necessary to refine treatment approaches and optimize patient outcomes.

## Figures and Tables

**Figure 1 diseases-13-00106-f001:**
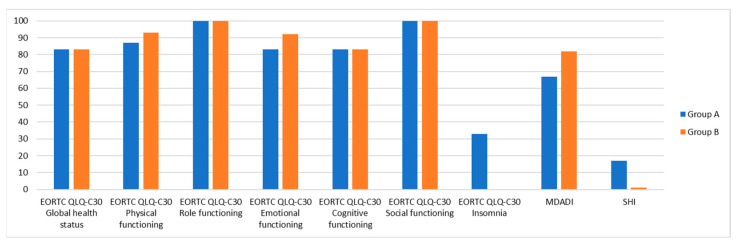
Multidimensional evaluation of OPSCC patients who underwent surgery (group A) or exclusive CRT (group B): EORTC QLQ-C30, MDADI, SHI. *p* values were >0.05 for every variable.

**Figure 2 diseases-13-00106-f002:**
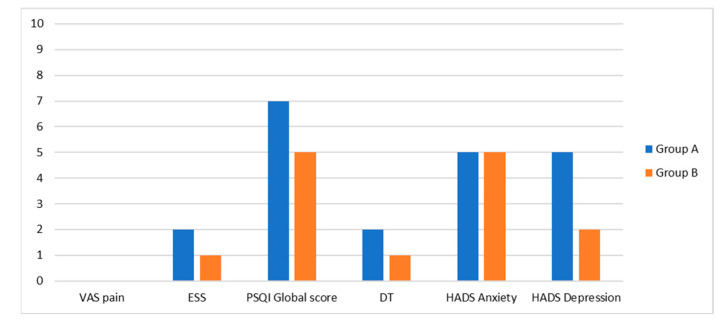
Multidimensional evaluation of OPSCC patients who underwent surgery (group A) or exclusive CRT (group B): VAS pain, ESS, PSQI, DT, HADS. *p* values were >0.05 for every variable.

**Table 1 diseases-13-00106-t001:** Clinical characteristics of both groups (*n* = 15 for each group).

Characteristics	Group A *n* (%)	Group B *n* (%)	*p* Value
Sex			0.705
Male	9 (60.0)	10 (66.7)	
Female	6 (40.0)	5 (33.3)	
Smoking			0.593
Never	3 (20.0)	3 (20.0)	
Former	11 (73.3)	12 (80.0)	
Active	1 (6.7)	0 (0.0)	
Alcohol assumption (>2 drinks for men, >1 drink for women)			0.283
Never	14 (93.3)	12 (80.0)	
Active	1 (6.7)	3 (20.0)	
Tumor site			0.179
Tonsillar lodge	7 (46.7)	8 (53.4)	
Tongue base	3 (20.0)	5 (33.3)	
Soft palate	4 (26.7)	0 (0.0)	
Posterior wall	1 (6.6)	2 (13.4)	
Tumor stage *			0.220
I	8 (53.4)	3 (20.0)	
II	2 (13.3)	2 (13.3)	
III	3 (20.0)	4 (26.7)	
IV	2 (13.4)	6 (40.0)	
HPV			0.705
Positive	9 (60.0)	10 (66.7)	
Negative	6 (40.0)	5 (33.3)	

* AJCC clinical staging system, 8th edition.

**Table 2 diseases-13-00106-t002:** Results from EORTC QLQ-C30 and EORTC H&N35 questionnaires in both groups (*n* = 15 for each group).

Scores	Group A Median (IQR)	Group B Median (IQR)	*p* Value
EORTC QLQ-C30			
Global health status	83.33 (33.3)	83.33 (8.33)	0.461
Physical functioning	86.67 (20.00)	93.33 (20.00)	0.567
Role functioning	100.00 (33.33)	100.00 (33.33)	0.683
Emotional functioning	83.33 (33.33)	91.67 (33.33)	0.870
Cognitive functioning	83.33 (16.67)	83.33 (16.67)	0.838
Social functioning	100.00 (33.33)	100.00 (33.33)	0.486
Fatigue	22.22 (33.33)	22.22 (33.33)	0.713
Nausea and vomiting	0.00 (0.00)	0.00 (0.00)	0.744
Pain	0.00 (16.67)	0.00 (33.33)	0.624
Dyspnea	0.00 (33.33)	0.00 (33.33)	0.935
Insomnia	33.33 (33.33)	0.00 (33.33)	0.217
Appetite loss	0.00 (33.33)	0.00 (0.00)	0.624
Constipation	0.00 (33.33)	0.00 (33.33)	0.935
Diarrhea	0.00 (0.00)	0.00 (0.00)	0.744
Financial difficulties	0.00 (0.00)	0.00 (66.67)	0.116
EORTC H&N35			
Pain in the mouth	0.00 (25.00)	8.33 (25.00)	0.806
Swallowing	8.33 (33.33)	8.33 (16.67)	0.902
Senses	0.00 (33.33)	0.00 (16.67)	0.567
Speech	22.22 (22.22)	0.00 (22.22)	0.567
Social eating	8.33 (16.67)	0.00 (16.67)	0.305
Social contact	0.00 (13.33)	0.00 (6.67)	0.389
Sexuality	0.00 (0.00)	0.00 (33.33)	0.250
Problems with teeth	0.00 (33.33)	0.00 (33.33)	0.870
Problems opening mouth	0.00 (66.67)	0.00 (33.33)	0.935
Dry mouth	0.00 (66.67)	33.33 (66.67)	0.567
Sticky saliva	33.33 (66.67)	33.33 (33.33)	0.683
Cough	0.00 (33.33)	0.00 (33.33)	0.436
Felt ill	0.00 (0.00)	0.00 (0.00)	0.512
Painkillers	0.00 (100.00)	0.00 (0.00)	0.775
Nutritional supplements	0.00 (100.00)	0.00 (0.00)	0.217
Feeding tube	0.00 (0.00)	0.00 (0.00)	1.000
Weight loss	0.00 (100.00)	0.00 (0.00)	0.367
Weight gain	0.00 (100.00)	0.00 (100.00)	1.000

EORTC, European Organization for Research and Treatment of Cancer; IQR, Interquartil Range.

**Table 3 diseases-13-00106-t003:** Results from ESS and PSQI questionnaires in both groups (*n* = 15 for each group).

Scores	Group A Median (IQR)	Group B Median (IQR)	*p* Value
ESS	2 (10)	1 (3)	0.233
PSQI			
Global score	7 (6)	5 (5)	0.325
Sleep quality	1 (0)	1 (0)	0.595
Sleep latency	1 (2)	1 (1)	0.539
Sleep duration	1 (3)	0 (1)	0.217
Habitual sleep efficiency	1 (3)	0 (3)	0.624
Sleep disturbances	1 (1)	1 (1)	0.595
Use of sleeping medications	0 (0)	0 (0)	1.000
Daytime dysfunction	1 (2)	1 (2)	0.595

ESS, Epworth Sleepiness Scale; IQR, Interquartil Range; PSQI, Pittsburgh Sleep Quality Index.

**Table 4 diseases-13-00106-t004:** Results from MDADI, SHI, DT and HADS questionnaires in both groups (*n* = 15 for each group).

Scores	Group A Median (IQR)	Group B Median (IQR)	*p* Value
MDADI	67 (21)	82 (18)	0.089
SHI	17 (36)	1 (7)	0.161
DT	2 (4)	1 (5)	0.870
HADS			
Anxiety	5 (4)	5 (4)	1.000
Depression	5 (7)	2 (5)	0.116

DT, Distress Thermometer; HADS, Hospital Anxiety Depression Scale; IQR, Interquartil Range; MDADI, MD Anderson Dysphagia Inventory; SHI, Speech Handicap Index.

## Data Availability

The data presented in this study are available on request from the corresponding author.

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
