# Peer review of "Surgery Versus Chemoradiation Therapy for Oropharyngeal Squamous Cell Carcinoma: A Multidimensional Cross-Sectional Study"

_diseases, 2025, doi:10.3390/diseases13040106_

Round 1
Reviewer 1 Report
Comments and Suggestions for Authors
In conclusion, while surgery and CRT are both effective in managing OPSCC, the authors conclude that both treatment modalities have comparable outcomes.
The topic is very interesting, and it is important that these studies are developed. It is well-structured and methodologically uses validated questionnaires. However, I believe that some aspects should be considered.
This study has some limitations. The small sample size reduced statistical power and limited the ability to detect subtle differences between groups. Assessments conducted over 12 months post-treatment overlooked early complications,
It is important to mention the histological type of the tumor, as some cancers respond better to surgery, while others are more sensitive to radiotherapy (RT).
Patient’s Condition Comorbidities: Other diseases of the patient should be considered as they may affect treatment tolerance.
Functional status (ECOG/Karnofsky Performance Status):
Surgery may require a prolonged recovery period.
RT may be a better option for patients who cannot undergo major surgery due to their overall condition.
Need for postoperative RT: Do some patients treated with surgery still require RT afterward?
Recurrence: Was there any recurrence in the evaluated patients, or was it completely absent?
Comments on the Quality of English LanguageSurgery and CRT are both effective in managing OPSCC, the authors conclude that both treatment modalities have comparable outcomes.
The topic is very interesting, and it is important that these studies are developed. It is well-structured and methodologically uses validated questionnaires. However, I believe that some aspects should be considered.
This study has some limitations. The small sample size reduced statistical power and limited the ability to detect subtle differences between groups. Assessments conducted over 12 months post-treatment overlooked early complications,
It is important to mention the histological type of the tumor, as some cancers respond better to surgery, while others are more sensitive to radiotherapy (RT).
Patient’s Condition Comorbidities: Other diseases of the patient should be considered as they may affect treatment tolerance.
Functional status (ECOG/Karnofsky Performance Status):
Surgery may require a prolonged recovery period.
RT may be a better option for patients who cannot undergo major surgery due to their overall condition.
Need for postoperative RT: Do some patients treated with surgery still require RT afterward?
Recurrence: Was there any recurrence in the evaluated patients, or was it completely absent?
Author Response
- Comments 1. In conclusion, while surgery and CRT are both effective in managing OPSCC, the authors conclude that both treatment modalities have comparable outcomes. The topic is very interesting, and it is important that these studies are developed. It is well-structured and methodologically uses validated questionnaires. However, I believe that some aspects should be considered.
- Response 1. Thank you for your positive comments.
- Comments 2. This study has some limitations. The small sample size reduced statistical power and limited the ability to detect subtle differences between groups. Assessments conducted over 12 months post-treatment overlooked early complications,
- Response 2. We discussed such limitations in the last paragraph of the Discussion section.
- Comments 3. It is important to mention the histological type of the tumor, as some cancers respond better to surgery, while others are more sensitive to radiotherapy (RT).
- Response 3. As mentioned in the article, the histological type was squamous cell carcinoma in all patients. The number and percentages of HPV-positive ones (that is, a prognostic factor for response to treatments) in both groups are reported in table 1.
- Comments 4. Patient’s Condition Comorbidities: Other diseases of the patient should be considered as they may affect treatment tolerance.
- Response 4. You are right, this is an important element so we added to the article the data about the mFI, modified frailty index, as it is an accurate measurement of the impact of cumulative comorbidities.
- Comments 5. Functional status (ECOG/Karnofsky Performance Status).
- Response 5. You are right, this is an important element and therefore we decided to add it in the revised version of the article.
- Comments 6. Surgery may require a prolonged recovery period.
- Response 6. Both surgery and chemoradiotherapy may require a prolonged recovery period. This is one of the reason for including patients with a follow-up >12 months. The absence of an analysis of early complications of both treatment modalities is discussed as a limitation of the study.
- Comments 7. RT may be a better option for patients who cannot undergo major surgery due to their overall condition.
- Response 7. We agree and we discuss it in the last paragraph of the Discussion section.
- Comments 8. Need for postoperative RT: Do some patients treated with surgery still require RT afterward?
- Response 8. The need for adjuvant postoperative RT is reported in the first paragraph of the Results section.
- Comments 9. Recurrence: Was there any recurrence in the evaluated patients, or was it completely absent?
- Response 9. Patients with recurrence or second tumors were excluded as reported in the Materials and methods section.
Reviewer 2 Report
Comments and Suggestions for Authors
Dear authors,
Thank you for the opportunity to review this article. In this article, you have conducted a cross-sectional study to analyze quality of life, speech, swallowing, sleep, psychological distress, and nutritional status in oropharyngeal squamous cell carcinoma (OPSCC) patients treated with either surgery or chemoradiotherapy (CRT). This multidimensional evaluation was performed by using validated questionnaires.
The study included 30 patients, 15 in each group. The patients included had a follow-up period longer that 12 months.
The manuscript is well written and presents valuable findings. Use of validated questionnaires for multidimensional evaluation of these two groups of patients ensures good assessment of patient outcome which is the strength of the paper. Also, this is an important topic in oncology because the decision of the treatment of patients with this diagnosis is not easy.
There are some limitations, but you have mentioned them, such as a small number of patients which limits statistical power of the results. Also, you have suggested future prospective studies comparing pre- and post-treatment outcomes, which I agree with.
Despite the limitations, the manuscript brings information important for treatment decisions. The paper is well-structured, methodologically sound, and gives a useful contribution to clinical decision-making in OPSCC management. I suggest acceptance.
Author Response
- Thank you for the opportunity to review this article. In this article, you have conducted a cross-sectional study to analyze quality of life, speech, swallowing, sleep, psychological distress, and nutritional status in oropharyngeal squamous cell carcinoma (OPSCC) patients treated with either surgery or chemoradiotherapy (CRT). This multidimensional evaluation was performed by using validated questionnaires. The study included 30 patients, 15 in each group. The patients included had a follow-up period longer that 12 months. The manuscript is well written and presents valuable findings. Use of validated questionnaires for multidimensional evaluation of these two groups of patients ensures good assessment of patient outcome which is the strength of the paper. Also, this is an important topic in oncology because the decision of the treatment of patients with this diagnosis is not easy. There are some limitations, but you have mentioned them, such as a small number of patients which limits statistical power of the results. Also, you have suggested future prospective studies comparing pre- and post-treatment outcomes, which I agree with. Despite the limitations, the manuscript brings information important for treatment decisions. The paper is well-structured, methodologically sound, and gives a useful contribution to clinical decision-making in OPSCC management. I suggest acceptance.
- Thank you very much for your positive comments.
Reviewer 3 Report
Comments and Suggestions for Authors
Thank you for the opportunity to review the manuscript "Surgery versus chemoradiation therapy for oropharyngeal squamous cell carcinoma: a multidimensional cross-sectional study", by Giuseppe Riva et al.
The manuscript is well designed and written in a clear and concise manner. The authors provided a consistent and informative introduction. The methodology was appropriate, and the results were well presented. The discussion, limitations and conclusions were outlined, compared and explored satisfactorily. Congratulations to the authors!
I recommend publishing this manuscript in Diseases.
Author Response
- Thank you for the opportunity to review the manuscript "Surgery versus chemoradiation therapy for oropharyngeal squamous cell carcinoma: a multidimensional cross-sectional study", by Giuseppe Riva et al. The manuscript is well designed and written in a clear and concise manner. The authors provided a consistent and informative introduction. The methodology was appropriate, and the results were well presented. The discussion, limitations and conclusions were outlined, compared and explored satisfactorily. Congratulations to the authors! I recommend publishing this manuscript in Diseases.
- Thank you very much for your positive comments.
Reviewer 4 Report
Comments and Suggestions for Authors
1. The introduction often states: "Patients with CRT often require
long-term gastrostomy tube dependence." This information needs to be
contained in a paragraph or expressed in paraphrase form.
2. 55% of the literature sources are old publications (more than 5 years
old). It is necessary to replace older items with newer ones.
3. In Table 1, the percentages need to be recalculated (rounding errors
resulted in results slightly greater than 100% or slightly less than 100%)
Author Response
- Comments 1. The introduction often states: "Patients with CRT often require long-term gastrostomy tube dependence." This information needs to be contained in a paragraph or expressed in paraphrase form.
- Response 1. You are right, there are some repetitions regarding the use of the gastrostomy tube; nevertheless, while this is an important aspect of rehabilitation, it is not the center point of our study and thus we decided not to dedicate a paragraph to this topic. We believe the need for assisted enteral feeding is an important aspect that we collaterally studied and reported in this article, but it should not have a center point in our introduction.
- Comments 2. 55% of the literature sources are old publications (more than 5 years old). It is necessary to replace older items with newer ones.
- Response 2. Thank you for pointing that out. We replaced three references with newer ones (ref n. 7: PMID 26461255 -> 39303189; ref n. 11: PMID 16027285 -> 39518095; ref n. 12: PMID 27688103 -> 39052674). Thanks to these substitutions, and excluding from the calculation the references necessary to report historically established questionnaries like the ones we used, we obtain a percentage of recent literatures (last 5 years) of 28%.
- Comments 3. In Table 1, the percentages need to be recalculated (rounding errors resulted in results slightly greater than 100% or slightly less than 100%)
- Response 3. The percentages in table 1 have been corrected.
Round 2
Reviewer 1 Report
Comments and Suggestions for Authors
The article is well written and addresses a relevant and timely topic. The proposed changes are appropriate and contribute to improving the clarity, of the manuscript.